# The Role of MicroRNAs in Uterine Leiomyosarcoma Diagnosis and Treatment

**DOI:** 10.3390/cancers15092420

**Published:** 2023-04-23

**Authors:** Iason Psilopatis, Kleio Vrettou, Stefania Kokkali, Stamatios Theocharis

**Affiliations:** 1First Department of Pathology, Medical School, National and Kapodistrian University of Athens, 75 Mikras Asias Street, Bld 10, Goudi, 11527 Athens, Greece; 2Department of Gynecology, Charité—Universitätsmedizin Berlin, Corporate Member of Freie Universität Berlin and Humboldt—Universität zu Berlin, Augustenburger Platz 1, 13353 Berlin, Germany; 3Oncology Unit, 2nd Department of Medicine, National and Kapodistrian University of Athens, Medical School, Hippocratio General Hospital of Athens, V. Sofias 114, 11527 Athens, Greece

**Keywords:** microRNA, uterine, sarcoma, biomarker, diagnosis, treatment

## Abstract

**Simple Summary:**

MicroRNAs (miRNAs) show differential expression in various cancer entities and seemingly contribute to cancer development and/or progression. Nevertheless, their role in uterine sarcoma diagnosis and treatment is poorly understood. The current work represents the most comprehensive, up-to-date review of the literature on the particular role of miRNAs as biomarkers for uterine sarcoma diagnosis and therapy.

**Abstract:**

Uterine sarcomas are rare gynecological tumors arising from the myometrium or the connective tissue of the endometrium with a relatively poor prognosis. MicroRNAs (miRNAs) represent small, single-stranded, non-coding RNA molecules that can function as oncogenes or tumor suppressors under certain conditions. The current review aims at studying the role of miRNAs in uterine sarcoma diagnosis and treatment. In order to identify relevant studies, a literature review was conducted using the MEDLINE and LIVIVO databases. The search terms “microRNA” and “uterine sarcoma” were employed, and we were able to identify 24 studies published between 2008 and 2022. The current manuscript represents the first comprehensive review of the literature focusing on the particular role of miRNAs as biomarkers for uterine sarcomas. miRNAs were found to exhibit differential expression in uterine sarcoma cell lines and interact with certain genes correlating with tumorigenesis and cancer progression, whereas selected miRNA isoforms seem to be either over- or under-expressed in uterine sarcoma samples compared to normal uteri or benign tumors. Furthermore, miRNA levels correlate with various clinical prognostic parameters in uterine sarcoma patients, whereas each uterine sarcoma subtype is characterized by a unique miRNA profile. In summary, miRNAs seemingly represent novel trustworthy biomarkers for the diagnosis and treatment of uterine sarcoma.

## 1. Introduction

Uterine sarcomas are highly malignant tumors arising from the smooth muscles and/or connective tissue elements of the uterus [1]. Depending on the type of cells they start in, uterine sarcomas may be categorized into four distinct classes. Uterine leiomyosarcomas represent the most common type and derive from the myometrium. Endometrial stromal sarcomas start in the endometrial stroma of the uterus and may be subdivided into low- and high-grade tumors, according to the cancer cell characteristics and tumor growth pattern. Undifferentiated sarcomas arise either from the endometrium or the myometrium and tend to grow and spread quickly. Adenosarcomas are biphasic neoplasms composed of benign epithelial elements and a malignant mesenchymal component which is usually low-grade, although high-grade sarcomatous overgrowth may occur [1].

Uterine sarcomas are relatively rare and represent 2–5% of all uterine cancers, with leiomyosarcomas and endometrial stromal sarcomas being the most common types [2]. The prevalence of uterine leiomyosarcomas in Afro-American women is twice as high as in Caucasian women [2]. Previous pelvic radiation therapy, tamoxifen intake, congenital retinoblastoma, as well as hereditary leiomyomatosis and renal cell cancer syndrome, are all linked to an increased risk of uterine sarcomas [3]. So far, specific chromosomal translocations have also been identified in diverse uterine sarcomas, with the resulting fusion genes leading to the activation of important transcription factors [4].

The signs and symptoms of uterine sarcomas are non-specific and may be mostly attributed to non-cancerous changes of the uterus, endometrial hyperplasia, or endometrial cancer. Abnormal bleeding or spotting, especially after menopause, is the most common symptom, followed by vaginal discharge, pain, feeling of a mass, and urine or bowel problems [5].

The diagnostic evaluation of uterine sarcomas includes, in addition to a physical examination, a transvaginal ultrasound and Magnetic Resonance Imaging (MRI), eventually combined with a Positron Emission Tomography (PET) scan. Nevertheless, definite diagnosis always requires hysteroscopic endometrial biopsy and tissue sampling in order to define the tumor grade and the hormone receptor status [6]. In their up-to-date review of the literature focusing on advances in the preoperative identification of uterine sarcoma, Liu et al. underlined the fact that high serum markers Cancer Antigen 125 (CA-125), Lactate Dehydrogenase (LDH), C-reactive protein (CRP), and D-dimers, could theoretically indicate uterine sarcoma, but are strongly influenced by various other factors and, consequently, lack specificity. Despite being a cheap and convenient screening method, ultrasound may not conclusively determine the benignity or malignancy of uterine masses. On the contrary, MRI exhibits great soft tissue resolution and certain degenerative types of uterine fibroids show comparable signal intensities that account for a certain rate of misdiagnosis. Last but not least, PET-CT is costly and hard to promote, but assures the highest accuracy [7].

For patients with early-stage resectable uterine leiomyosarcoma and undifferentiated sarcoma, hysterectomy, along with bilateral salpingo-oophorectomy, represent the mainstay of treatment. In cases where sarcoma recurrence is highly expected, adjuvant radiochemotherapy might complete the treatment plan. Patients with advanced disease are mainly treated with systemic therapy, especially when complete surgical excision is impossible. The treatment concept is similar for endometrial stromal sarcomas, with the addition of hormonal therapy in cases of positive hormone receptor status [8]. In this context, Bose et al. recently published their comprehensive article on novel therapeutics in the treatment of uterine sarcomas and highlighted that, for women with advanced uterine leiomyosarcomas, the targeting of DNA damage repair pathways, alongside a depletion of immunosuppressive macrophage populations, could undoubtedly represent promising therapeutic approaches. Of note, several endometrial stromal sarcomas are even characterized by potentially actionable modifications in the Wingless/Integrate (Wnt), cyclin D-cyclin-dependent kinase (CDK), 4/6-Retinoblastoma (Rb), and murine double minute 2 (mdm2)–p53 pathways [9].

Polo-like kinase 4 (PLK4), Secreted Protein, Acidic and Rich in Cysteine (SPARC) Related Modular Calcium Binding 2 (SMOC2), Special AT-rich Sequence-Binding protein 2 (SATB2), or Forkhead box P3 (FOXP3) + T cells have all been suggested as prognostic indicators for uterine sarcomas [10]. More accurately, PLK4 has been proposed to represent a key regulator of centriole replication, which correlates with the following five hallmarks of cancer: proliferative signaling sustainment, tumor-promoting inflammation, invasion and metastasis activation, genome instability and mutation, as well as resistance to cell death [11]. Furthermore, PLK4 expression levels have been reported to be significantly higher in malignant tumor samples, with this overexpression constituting a biomarker predicting meagre prognosis for many human cancers [11]. SMOC2 is a matricellular protein which evidently not only influences cell migration, adhesion, and tissue repair, but also exhibits tumor suppressor capacities in cancer advancement, thereby embodying a promising prognostic marker [12]. Moreover, SATB2 is a transcriptional co-factor that controls chromatin architecture so as to modulate gene expression that regulates pluripotency and self-renewal [13]. According to The Cancer Genome Atlas (TCGA) expression data, SATB2, alone or combined with other proteins, could potentially be employed as a useful biomarker for cancer diagnosis [13]. Last but not least, FOXP3+ regulatory T cells play a crucial role in the maintenance of immune tolerance, alongside homeostasis of the immune system [14]. As a prognostic indicator, FOXP3+ regulatory T cells seem to negatively affect OS depending on each tumor site, the molecular subtypes, and tumor stages [14].

MicroRNAs (miRNAs) are small non-coding RNAs, with an average length of 22 nucleotides [15]. DNA sequences are first transcribed into primary miRNAs, which are then processed into precursor and mature miRNAs. miRNAs are best known to interact with the 3′ UTR of target messenger RNAs (mRNAs) to downregulate expression [16]. Under adequate circumstances, miRNAs may even activate gene expression or control the rate of translation and transcription [17,18,19]. Alternative splicing and polyadenylation of 3′ UTR, along with cell-type-specific RNA binding proteins affecting target mRNA secondary structures, influence mRNA sensitivity to miRNA-driven gene regulation in a cell type/state-specific way [15].

Upon release into extracellular fluids, miRNAs may reach target cells and act as autocrine, paracrine, and/or endocrine regulators to moderate cellular pathways [20]. By acting as intercellular signaling molecules, miRNAs may promote cell proliferation and migration [21,22], induce angiogenesis [22], activate downstream signaling events [23], as well as lead to biological responses and neurodegeneration [24]. Unlike intracellular RNAs, extracellular miRNAs show high stability and resistance to degradation in deleterious conditions [25]. Their protection in the extracellular milieu may be mainly attributed to the presence of miRNAs in vesicles, such as exosomes, microvesicles, and apoptotic bodies, or with accompanying proteins, especially Argonaute RISC Catalytic Component 2 (AGO2) [26]. Vesicle-associated extracellular miRNAs seem to employ endocytosis, phagocytosis, or direct fusion with the plasma membranes to enter cells, whereas vesicle-free secreted miRNAs take advantage of specific receptors on the cell surface [27]. The release of extracellular miRNAs is regulated by various pathways, including a ceramide-dependent pathway [28], vesicle-associated membrane protein 3 (VAMP3) and synaptosomal-associated protein 23 (SNAP23) [29], the soluble N-ethylmaleimide-sensitive factor activating protein receptor (SNARE) complex [30], as well as different signaling molecules [31,32].

MiRNAs determine normal animal development and play a significant role in development, differentiation, proliferation, apoptosis, and immune responses [33]. Nevertheless, aberrant miRNA expression correlates with a wide range of diseases, including diabetes, cardiovascular disease, kidney disease, and cancer [34]. Given their secretion into extracellular fluids, extracellular miRNAs have been widely reported as potential biomarkers. A recent comprehensive review of the literature presented a list of circulating miRNA isoforms that are most commonly associated with ten cancer diseases, ranging from esophageal or gastric cancer to melanoma and breast cancer [35]. Furthermore, Duica et al. pointed out the great potential of miRNAs in deciphering gynecological malignancies [36].

To our knowledge, no review article has, to date, been published on the role of miRNAs in the diagnosis and treatment of uterine sarcoma. The present work aims to investigate the usage of miRNAs as potential biomarkers in uterine sarcoma. The literature review was conducted using the MEDLINE and LIVIVO databases. Solely original research articles and scientific abstracts written in the English language that explicitly reported on miRNAs in uterine sarcoma were included in the data analysis. Studies that incorporated solely uterine leiomyomas or carcinosarcomas, as well as studies not clearly stating the exact histological tumor characteristics (e.g., uterine leiomyosarcoma) were excluded. The search terms ‘‘microRNA’’ and ‘‘uterine sarcoma’’ were employed, and we were able to identify a total of 37 articles published between 2008 and 2022, after the exclusion of duplicates. A total of 10 articles were discarded in the initial selection process after abstract review. The full texts of the remaining 27 publications were evaluated, and after detailed analysis, a total of 24 relevant studies published between 2008 and 2022 met the inclusion criteria and were selected for the literature review. Figure 1 presents an overview of the aforementioned selection process.

## 2. The Expression of miRNAs in Uterine Sarcoma

### 2.1. Differential Expression of miRNAs in Uterine Leiomyosarcoma Cell Lines

To date, a great number of studies have investigated the role of miRNAs in the diagnosis and treatment of uterine leiomyosarcoma.

In 2012, Chuang et al. studied the human leiomyosarcoma cell line SKLM-S1 and published two original research articles on the role of miRNAs in leiomyosarcomas [37,38]. More precisely, the gain of function of miRNA-93/106b in SKLM-S1 cells revealed that these miRNA isoforms regulated the expression of both *F3* and *interleukin 8* (*IL8*) as their direct targets, as well as the expression of *connective tissue growth factor* (*CTGF*) and *plasminogen activator inhibitor 1* (*PAI1*) as their indirect targets through tissue factor (F3)-mediated signaling [37]. Similarly, the gain of function of miRNA-200c in SKLM-S1 cells confirmed *zinc-finger E-box binding homeobox 1/2* (*ZEB1/2*) and *vascular endothelial growth factor-A* (*VEGFA*), and validated *fibulin 5* (*FBLN5*) and *tissue inhibitor of metalloproteinases 2* (*TIMP2*), as direct miRNA-200c targets through interaction with their respective 3′ UTRs [38]. Three years later, the same study group used the SK-LMS-I in vitro model and, by employing a quantitative polymerase chain reaction and Western blot analysis, suggested that miRNA-200c gain of function suppresses *inhibitor of kappa light polypeptide gene enhancer in B-cells, kinase beta* (*IKBKB*), *IL8*, *CDK2*, and *cyclin E2* (*CCNE2*) expression, decreases p65 transcriptional activity in the *IL8* promoter, increases SK-LMS-1 caspase 3/7 activity, as well as inhibits their proliferation and migration [39]. Shi et al. examined the translational regulation of *high-mobility-group AT-hook 2* (*HMGA2*) by *lethal-7* (*let-7*) and outlined that the *let-7*-mediated *HMGA2* repression in uterine leiomyosarcoma cell lines led to in vitro uterine leiomyosarcoma cell growth inhibition [40]. Additionally, Yang et al. demonstrated that miRNAs may regulate gene expression in uterine leiomyosarcoma cells in response to bromodomain 9 (BRD9) inhibition, given that the genes induced by TP-472 treatment correlated with miRNA-4776-5p, miRNA-671-3p, miRNA-3619-3p, miRNA-621, and miRNA-553, whereas genes suppressed by TP-472 treatment were associated with miRNA-542-5p, miRNA-4734, miRNA-3682-3p, as well as miRNA-4727-3p [41]. Altogether, miRNAs seem to show differential expression in uterine leiomyosarcoma cell lines and interact with genes influencing tumor development and progression (Table 1).

### 2.2. Different miRNA Profiles between Uterine Leiomyosarcomas versus Normal Uteri and Benign Uterine Tumors

Benna et al. employed the Sarcoma miRNA Expression Database for the comparison of leiomyosarcoma and smooth muscle samples, in terms of differential miRNA expression. The expression of 301 out of the 1120 miRNAs tested was found to significantly differ between leiomyosarcoma and smooth muscle samples, with 172 of these identified miRNAs targeting a total of 438 genes involved in specific molecular pathways. Most importantly, pathway analysis revealed the involvement of RNA Polymerase III, transfer RNA (tRNA) functions, as well as dopamine-mediated synaptic neurotransmission in leiomyosarcoma development [42]. Furthermore, Danielson et al. hybridized ten frozen samples of uterine leiomyosarcoma to Agilent arrays and found 32 upregulated and 40 downregulated miRNAs, whereas unsupervised hierarchical clustering revealed that the miRNA profile may accurately cluster normal myometrium, benign tumors, and uterine leiomyosarcoma, based on the distance along uterine smooth muscle differentiation. Of note, time progression and phylogenetic analyses based on miRNA expression profiles clustered uterine leiomyosarcomas with human mesenchymal stem cells [43]. Kowalewska et al. assessed 88 miRNAs by quantitative real-time polymerase chain reaction (qRT-PCR) in normal uteri and cancerous samples from patients with leiomyosarcoma, endometrial sarcoma, and mixed epithelial–mesenchymal tumors and highlighted that *miRNA-23b*, *miRNA-1*, *let-7f*, and *let-7c*, are downregulated in endometrial sarcomas, whereas there are no statistically significant changes in miRNA expression levels between leiomyosarcomas and normal uteri [44]. Moreover, Nuovo et al. studied miRNA expression in 15 leiomyosarcomas and detected high miRNA-221 expression levels by in situ hybridization in 13 out of 15 cases. On the contrary, miRNA-221 was not expressed in leiomyoma nor in benign metastasizing leiomyoma patients [45]. Renner et al. collected 13 leiomyosarcomas for miRNA profiling and found 28 upregulated miRNAs (including the muscle-specific *myomiRNAs miRNA-133a*, *miRNA-133b*, and *miRNA-1*) and 13 downregulated miRNAs. Interestingly, 10 out of 13 leiomyosarcoma samples clustered to the subgroup with decreased 14q32.2 miRNA expression [46]. Even though *miRNA-1* was found to be strongly suppressed in uterine leiomyosarcoma tissue samples, Stope et al. failed to detect the growth inhibitory capacities of *miRNA-1* in the SK-UT-1 cell line, with *miRNA-1* not affecting the expression of the cell survival and mitogen-activated protein (MAP) kinases extracellular signal-regulated kinase 1/2 (ERK1/2) and p38 [47]. Yokoi et al. detected the downregulation of miRNA-4430, miRNA-6511b-5p, miRNA-451a, miRNA-4485-5p, miRNA-4635, miRNA-1246, and miRNA-191-5p in uterine leiomyosarcoma serum samples and pointed out the supremacy of miRNA-1246 and miRNA-191-5p for uterine leiomyosarcoma diagnosis, with an area under the receiver operating characteristic curve (AUC) of 0.97 (95% confidence interval (CI), 0.91–1.00) [48]. Additionally, Ventura et al. analyzed uterine tissue samples and described a trend toward miRNA-126 hypo-expression between uterine leiomyoma and benign metastasizing leiomyoma, whereas higher miRNA-221 and lower miRNA-126 expression was observed in leiomyosarcomas [49]. Taken together, selected miRNA isoforms seem to be either upregulated or downregulated in uterine leiomyosarcoma samples compared to normal uteri or benign tumors, constituting a significantly different miRNA profile for malignant tumor entities (Table 2).

### 2.3. Other Smooth Muscle Neoplasms versus Leiomyosarcoma

In addition to leiomyosarcoma, smooth muscle neoplasms of the uterus include leiomyoma (with variants such as cellular leiomyoma, mitotically active leiomyoma, and atypical or bizarre leiomyoma) and smooth muscle tumor of uncertain malignant potential (STUMP). Zhang et al. analyzed a cohort of 167 uterine smooth muscle tumors, including 38 leiomyosarcomas, 18 STUMPs, 42 bizarre leiomyomas, 22 cellular leiomyomas, 7 mitotically active leiomyomas, and 40 conventional leiomyomas. They found that bizarre leiomyomas and leiomyosarcomas share similar miRNA signatures [50].

### 2.4. Endometrial Stromal Sarcoma versus Leiomyosarcoma

Endometrial stromal sarcomas start in the endometrial uterine stroma and, together with leiomyosarcomas, represent the most common histological types of uterine sarcoma [51].

Ravid et al. compared the miRNA profiles of uterine endometrial stromal sarcoma and leiomyosarcoma, alongside the miRNA signatures of primary and metastatic uterine leiomyosarcoma, and pointed out that 76 miRNAs were overexpressed in endometrial stromal sarcoma, 18 in leiomyosarcoma, 45 in primary leiomyosarcoma, and 4 in metastases. Using short-interfering RNA (siRNA), frizzled-6 was silenced in the SK-LMS-1 uterine leiomyosarcoma cell line, thus significantly inhibiting cellular invasion, wound healing, and matrix metalloproteinase-2 (MMP-2) activity [51].

### 2.5. Undifferentiated Pleomorphic Sarcoma versus Leiomyosarcoma

Undifferentiated pleomorphic sarcoma is an adult soft tissue sarcoma and an exclusion diagnosis without any specific known differentiation lineages and with (myo-)fibroblastic or smooth-muscle-like spindle cell components [52].

Guled et al. conducted miRNA profiling on 37 leiomyosarcoma and 31 undifferentiated pleomorphic sarcoma samples and reported differential expression for miRNA-199b-5p, miRNA-320a, miRNA-199a-3p, miRNA-126, as well as miRNA-22. To be more accurate, miRNA-199b-5p was highly expressed in undifferentiated pleomorphic sarcomas, whereas leiomyosarcomas were associated with high miRNA-320a levels. Notably, miRNA-22 seemed to have the leiomyosarcoma-associated *receptor tyrosine kinase (RTR)-like orphan receptor 2* (*ROR2*) as its target gene [52].

Taken together, each uterine sarcoma class seems to be characterized by a unique miRNA profile.

### 2.6. miRNAs Correlate with Prognosis of Uterine Leiomyosarcomas and Could Predict Response to Treatment

De Almeida et al. cultured the uterine leiomyosarcoma SK-UT-1 HTB-114 cell line and, after performing qRT-PCR, identified five upregulated and eight downregulated miRNAs. Specifically, miRNA-1-3p, miRNA-202-3p, and miRNA-7-5p, which presented a similar expression pattern in the SK-UT-1 HTB-114 cell line in comparison with 16 formalin-fixed paraffin-embedded uterine leiomyosarcoma samples, showed significant expression in uterine leiomyosarcoma [53]. Two years later, the same study group assessed the miRNA expression profile in 34 leiomyosarcoma paraffin-embedded samples and reported downregulation of all the *let-7* miRNA group members. Decreased *let-7e* expression was associated with a worse overall survival (OS) and a high distant metastasis rate, whereas patients with low *let-7b* and *let-7d* levels showed worse disease-free survival (DFS). As for the patients’ age, older patients had the lowest *let-7d*, *let-7e*, and *let-7f* expression levels [54]. Gonzalez dos Anjos et al. selected 37 uterine leiomyosarcoma formalin-fixed paraffin-embedded samples and underlined the association of lower cancer-specific survival (CSS) with the upregulation of miRNA-196a-5p and miRNA-34c-5p, as well as the downregulation of miRNA-125a-5p and miRNA-10a-5p. In 18 endometrial stromal sarcoma formalin-fixed paraffin-embedded samples, the overexpression of miRNA-373-3p, miRNA-372-3p, and let-7b-5p, along with the decrease of let-7f-5p, miRNA-23-3p, and let-7b-5p, correlated with a lower CSS, respectively. High miRNA-138-5p levels were associated with better survival. Furthermore, miRNA-335-5p, miRNA-301a-3p, and miRNA-210-3p correlated with uterine sarcoma metastasis and relapse, whereas miRNA-138-5p, miRNA-146b-5p, and miRNA-218-5p expression predicated higher DFS in treated patients [55]. Additionally, Schiavon et al. performed RT-PCR analyses in 37 formalin-fixed paraffin-embedded uterine leiomyosarcoma tissue samples and reported the downregulation of 19 miRNAs and the upregulation of 25 miRNAs. More precisely, *miRNA-148a-3p* significantly correlated with tumor relapse, *miRNA-27b-3p* with metastasis, and *miRNA-124-3p* and *miRNA-183-5p* with patient death, whereas low *miRNA135b-5p* levels were associated with DFS. In comparison with normal myometrium or benign leiomyomas, *miRNA144-3p*, *miRNA34a-5p*, and *miRNA206* constituted a signature for the distinction of uterine leiomyomas from leiomyosarcomas [56]. The Cancer Genome Atlas Research Network discovered that 12 miRNAs correlated with recurrence-free survival (RFS) in leiomyosarcoma, with miRNA-181b-5p representing the miRNA with the highest association with RFS and high miRNA-181b being more common in uterine than in soft tissue leiomyosarcoma [57]. Tong et al. enrolled 101 patients with uterine sarcoma and measured the levels of different serum miRNAs by qRT-PCR. miRNA-152 and miRNA-24 were downregulated, whereas miRNA-205, miRNA-222, and miRNA-150 were upregulated. Furthermore, all the miRNAs correlated with the sarcoma stage and uterine sarcoma patients with high-level miRNA-152 and miRNA-24 exhibited significantly better survival rates [58]. Wiemer et al. explored the role of microRNAs in terms of eribulin sensitivity or resistance in sarcomas and presented statistically significant differences in the expression of miRNA-1271, miRNA-146a, let-7g, miRNA-574-3p, miRNA-362-3p, miRNA-181a-2, miRNA-29b-2, and miRNA-590-3p between eribulin responders and non-responders for leiomyosarcomas [59]. Eribulin is a marine-derived drug, a structurally modified analog of halichondrin B, which is isolated from the sponge *Halichondria okadai* and has clinical activity mainly against liposarcomas and to a lesser extent leiomyosarcoma [60,61]. In other cancers, the expression of specific miRNAs has been associated with drug resistance [62]. Accurate biomarkers that predict the response to a certain therapy are valuable tools for personalized treatment in clinical practice. Altogether, miRNA overexpression or depletion seem to correlate with various clinical prognostic parameters in uterine leiomyosarcoma patients (Table 3).

## 3. Discussion

Uterine sarcomas are rare gynecologic tumors with a high degree of malignancy and a relatively poor prognosis. Although low grade endometrial stromal sarcomas are associated with a five-year relative survival rate of 78%. Even in a distant Surveillance, Epidemiology, and End Results (SEER) stage, the five-year relative survival rates for high grade endometrial stromal sarcomas resemble those for undifferentiated sarcomas. Leiomyosarcomas are the subgroup with the worst prognosis, given that the five-year relative survival rate amounts to only 39% for all the SEER stages combined [63]. Importantly, each uterine sarcoma histologic subtype also shows a unique clinical course, may only be accurately diagnosed postoperatively, and can be challenging to differentiate from similar benign lesions [64]. As such, uterine sarcomas still represent a diagnostic and therapeutic challenge that seemingly requires our better understanding of immunophenotypes and molecular characterization. In the present review of the literature, we highlight the role of miRNAs as novel biomarkers in terms of the diagnosis and treatment of uterine sarcoma. To our knowledge, the current work represents the most up-to-date comprehensive literature review on this topic and includes a total of 24 relevant original research articles.

By carefully analyzing the aforementioned articles, firstly, we conclude that miRNAs exhibit differential expression in uterine leiomyosarcoma cell lines and interact with certain genes that are (partly) responsible for tumorigenesis and cancer progression. This observation is very interesting, especially in the context of personalized/targeted medicine, as the involved miRNAs might act as useful targets of novel treatment agents ranging from miRNA mimics to overexpress the transcript, to miRNA repressors to silence transcript function. Of note, alleged miRNA drugs have so far displayed significant efficacy in various health conditions, such as cancer, hepatitis C, heart abnormalities, and kidney failure [62].

Secondly, selected miRNA isoforms show either elevated or decreased expression in uterine leiomyosarcoma samples compared to normal uteri or benign tumors. Consecutively, miRNAs might preoperatively give the clinicians the opportunity to determine the malignancy of a suspicious uterine mass and hopefully help avoid unnecessary and complicated surgical procedures.

Thirdly, miRNA overexpression or depletion is associated with various clinical prognostic parameters in uterine leiomyosarcoma patients. More precisely, miRNA expression levels significantly correlate with OS, DFS, metastasis rate, or tumor relapse, all of which are of utmost importance for accurate patient education. This prognostic information could be used to make adjuvant therapy (chemotherapy, radiation therapy) decisions. The role of these therapies is not well established in localized uterine leiomyosarcomas. Future clinical trials exploring the potential benefit of adjuvant treatment based on biomarkers, such as miRNAs, could at least partially reverse the poor prognosis of distinct uterine sarcoma histotypes. Interestingly, a study in patients with osteosarcoma is currently investigating potential prognostic and predictive tumor tissue and blood biomarkers, including miRNAs (NCT01190943). Another study is currently assessing plasma miRNAs as potential biomarkers to guide the decision for lymphadenectomy in patients with endometrial cancer (NCT03776630). Studies on miRNAs expression in other cancers, such as breast and lung cancer, have demonstrated an association with the efficacy of curative surgical treatment [35].

Even among the different uterine sarcoma subgroups, each uterine sarcoma class is characterized by a unique miRNA profile. This finding is undoubtedly promising because, as mentioned above, low grade endometrial stromal sarcomas, for instance, have a better prognosis or require a much less complicated treatment plan than leiomyosarcomas [63]. The different miRNAs that regulate gene expression in uterine leiomyosarcoma cell lines, show differential expression in uterine leiomyosarcomas vs. normal uteri and benign uterine tumors, or correlate with clinical prognostic parameters in uterine leiomyosarcoma patients are comprehensively summarized in Table 1, Table 2 and Table 3. *miRNA-23b*, *miRNA-1*, *let-7f*, and *let-7c* are downregulated in endometrial sarcomas, whereas the overexpression of miRNA-373-3p, miRNA-372-3p, and let-7b-5p, along with a decrease in let-7f-5p, miRNA-23-3p, and let-7b-5p, correlate with a lower CSS. miRNA-199b-5p seems to be overexpressed in undifferentiated pleomorphic sarcomas.

Nevertheless, despite the numerous advantages, several shortcomings of miRNAs as new biomarkers in the diagnosis and treatment of uterine sarcoma need to be discussed. The experimental determination of the miRNA–mRNA interactions is both costly and time-consuming, whereas miRNA–mRNA base pairing in mammals is not always necessarily complementary, thereby rendering the accurate prediction of miRNA targets rather demanding [65]. Besides, our limited experience with miRNAs justifies the absence of well-established detection limits, concentration levels in body fluids, or the parameters modulating miRNA expression [66]. Another important drawback is the fact that the use of venous miRNAs for cancer detection might be challenged in certain cases [67]. Last but not least, several identified miRNAs are characterized by relatively poor diagnostic specificity and reproducibility and, therefore, require novel standardized detection techniques [68].

## 4. Conclusions

In summary, miRNAs might successfully represent novel trustworthy biomarkers in the field of uterine sarcoma diagnosis and treatment. However, further clinical studies in larger patient collectives need to be conducted in order to verify the clinical utility of miRNAs in the diagnosis and treatment of uterine sarcoma, as well as to clarify contradictory results that arise from small, heterogenous patient cohorts [46,47].

## Figures and Tables

**Figure 1 cancers-15-02420-f001:**
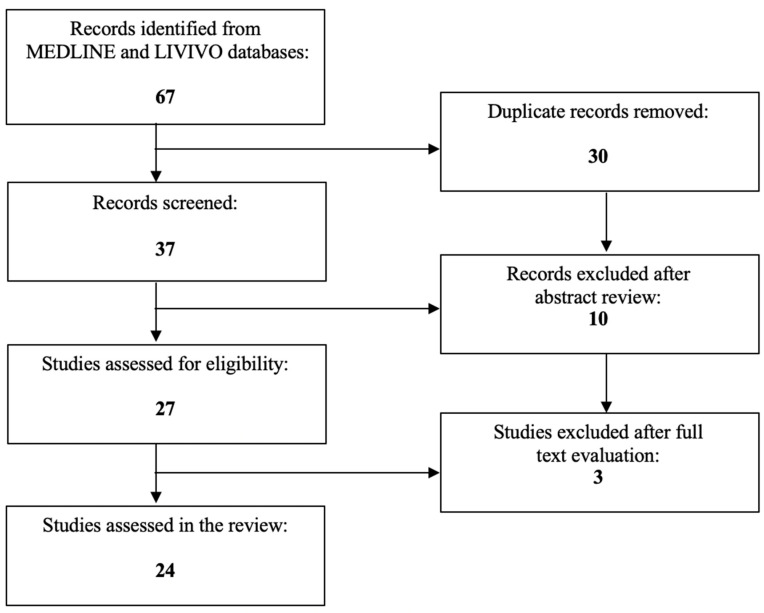
PRISMA flow diagram visually summarizing the screening process.

**Table 1 cancers-15-02420-t001:** miRNA-mediated gene expression regulation in uterine leiomyosarcoma cell lines.

miRNA	Expression Level	Associated Gene	Reference
miRNA-93/106b	Gain of function	*F3*	[37]
		*IL8*	
		*CTGF*	
		*PAI1*	
miRNA-200c	Gain of function	*ZEB1/2*	[38,39]
		*VEGFA*	
		*FBLN5*	
		*TIMP2*	
		*IKBKB*	
		*IL8*	
		*CDK2*	
		*CCNE2*	
let-7	Upregulation	*HMGA2*	[40]
miRNA-4776-5p	Upregulation or downregulation	Genes linked to BRD9 inhibition	[41]
miRNA-671-3p			
miRNA-3619-3p			
miRNA-621			
miRNA-553			
miRNA-542-5p			
miRNA-4734			
miRNA-3682-3p			
miRNA-4727-3p			

**Table 2 cancers-15-02420-t002:** Differentially expressed miRNAs in uterine leiomyosarcomas vs. normal uteri and benign uterine tumors.

miRNA	Differential Expression	Reference
miRNA-23b	Downregulation in endometrial sarcomas No statistically significant changes between leiomyosarcomas and normal uteri	[44]
miRNA-1		
let-7f		
let-7c		
miRNA-221	Upregulation in leiomyosarcoma Absence in leiomyoma and benign metastasizing leiomyoma	[45]
miRNA-133a	Upregulation in leiomyosarcoma	[46]
miRNA-133b		
miRNA-1		
miRNA-1	Strong suppression in leiomyosarcoma	[47]
miRNA-4430	Downregulation in leiomyosarcoma	[48]
miRNA-6511b-5p		
miRNA-451a		
miRNA-4485-5p		
miRNA-4635		
miRNA-1246		
miRNA-191-5p		
miRNA-126	Downregulation in leiomyosarcoma Hypo-expression between uterine leiomyoma and benign metastasizing leiomyoma	[49]
miRNA-221	Upregulation in leiomyosarcoma	[49]

**Table 3 cancers-15-02420-t003:** miRNA isoforms associated with clinical prognostic parameters in uterine leiomyosarcoma patients.

miRNA	Clinical Parameter	Reference
let-7b	OS DFS Distant metastasis rate	[54]
let-7d		
let-7e		
let-7f		
miRNA-196a-5p	CCS DFS Metastasis Tumor relapse	[55]
miRNA-34c-5p		
miRNA-125a-5p		
miRNA-10a-5p		
miRNA-373-3p		
miRNA-372-3p		
let-7b-5p		
let-7f-5p		
miRNA-23-3p		
let-7b-5p		
miRNA-138-5p		
miRNA-335-5p		
miRNA-301a-3p		
miRNA-210-3p		
miRNA-146b-5p		
miRNA-218-5p		
miRNA-148a-3p	Tumor relapse Metastasis Patient death DFS	[56]
miRNA-27b-3p		
miRNA-124-3p		
miRNA-183-5p		
miRNA135b-5p		
miRNA-181b-5p	RFS	[57]
miRNA-152	Tumor stage Patient survival	[58]
miRNA-24		
miRNA-205		
miRNA-222		
miRNA-150		
miRNA-1271	Response to eribulin	[59]
miRNA-146a		
let-7g		
miRNA-574-3p		
miRNA-362-3p		
miRNA-181a-2		
miRNA-29b-2		
miRNA-590-3p		

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
