# Peer review of "The Role of MicroRNAs in Uterine Leiomyosarcoma Diagnosis and Treatment"

_cancers, 2023, doi:10.3390/cancers15092420_

Round 1
Reviewer 1 Report
This article by Iason Psilopatis et al. mainly discussed the role of miRNA in the diagnosis and treatment of uterine sarcoma. This is a comprehensive review of the specific role of miRNA as biomarkers of uterine sarcoma. It was found that multiple miRNAs were differentially expressed in uterine sarcoma cell lines with a correlation between the miRNA levels and clinical prognostic parameters of patients bearing uterine sarcoma, suggesting that miRNA is a novel and reliable biomarker for the diagnosis and treatment of uterine sarcoma. It is educative and full of significances. It is therefore recommended to be published with minor revisions.
The suggestions are as follows:
1. The title of the second part of this paper is "the role of miRNA in the diagnosis and treatment of uterine sarcoma", but it actually mainly talks about the expression of miRNA in uterine sarcoma, and the role is scarcely mentioned, as well as the expression and application in treatment. It is suggested to add more discussions or modify the title to be appropriate.
2. In the third part of the paper, some other cancer marker molecules seem to have little relationship with the miRNA focused in this paper, which is rather inconsistent. This paragraph can be put into the background part.
3. The authors should have some discussions about shortcomings of miRNA as a new biomarker in the diagnosis and treatment of uterine sarcoma, so as to make the paper unbiased.
Moderate editing of English language
Author Response
This article by Iason Psilopatis et al. mainly discussed the role of miRNA in the diagnosis and treatment of uterine sarcoma. This is a comprehensive review of the specific role of miRNA as biomarkers of uterine sarcoma. It was found that multiple miRNAs were differentially expressed in uterine sarcoma cell lines with a correlation between the miRNA levels and clinical prognostic parameters of patients bearing uterine sarcoma, suggesting that miRNA is a novel and reliable biomarker for the diagnosis and treatment of uterine sarcoma. It is educative and full of significances. It is therefore recommended to be published with minor revisions.
The suggestions are as follows:
- The title of the second part of this paper is "the role of miRNA in the diagnosis and treatment of uterine sarcoma", but it actually mainly talks about the expression of miRNA in uterine sarcoma, and the role is scarcely mentioned, as well as the expression and application in treatment. It is suggested to add more discussions or modify the title to be appropriate.
We have now changed the title to ‘’The expression of miRNAs in uterine sarcoma’’, as suggested by the reviewer.
- In the third part of the paper, some other cancer marker molecules seem to have little relationship with the miRNA focused in this paper, which is rather inconsistent. This paragraph can be put into the background part.
Following the reviewer’s suggestion, we have now put the named paragraph into the background part.
- The authors should have some discussions about shortcomings of miRNA as a new biomarker in the diagnosis and treatment of uterine sarcoma, so as to make the paper unbiased.
Thank you for this useful comment. We have now added some discussions about shortcomings of miRNA as a new biomarker in the diagnosis and treatment of uterine sarcoma.
Reviewer 2 Report
This review article aims to describe the role of miRNAs in uterine sarcoma diagnosis and treatment. The following revisions are recommended:
- Adenosarcoma is a biphasic neoplasm rather than a pure uterine sarcoma. I am not sure it should be included in this review article on uterine sarcomas at all, but if it is included, its definition should be clearer on page 1, line 40 - "Adenosarcomas are generally low-grade cancers with normal gland cells mixed with cancer cells of the supporting uterine connective tissue" should change to "Adenosarcomas are biphasic neoplasms composed of benign epithelial elements and a malignant mesenchymal component which is usually low-grade, although high-grade sarcomatous overgrowth may occur."
- The manuscript consists of long paragraphs with description of expression of different miRNAs in uterine sarcomas. The manuscript needs to clearly describe miRNA expression profiles separately for each type of uterine sarcoma - leiomyosarcoma, low-grade endometrial stromal sarcoma, high-grade endometrial stromal sarcoma (different subtypes if data is available), and undifferentiated uterine sarcoma. Again, adenosarcoma can be either removed or included as a separate entity - any relationship of miRNA expression with sarcomatous overgrowth and/or high-grade morphology? Also, for each sarcoma subtype it should be clearly described which miRNAs can be used for diagnosis and which are prognostic/predictive.
See above
Author Response
This review article aims to describe the role of miRNAs in uterine sarcoma diagnosis and treatment. The following revisions are recommended:
- Adenosarcoma is a biphasic neoplasm rather than a pure uterine sarcoma. I am not sure it should be included in this review article on uterine sarcomas at all, but if it is included, its definition should be clearer on page 1, line 40 - "Adenosarcomas are generally low-grade cancers with normal gland cells mixed with cancer cells of the supporting uterine connective tissue" should change to "Adenosarcomas are biphasic neoplasms composed of benign epithelial elements and a malignant mesenchymal component which is usually low-grade, although high-grade sarcomatous overgrowth may occur."
We have now modified the definition of adenosarcoma, as suggested by the reviewer.
- The manuscript consists of long paragraphs with description of expression of different miRNAs in uterine sarcomas. The manuscript needs to clearly describe miRNA expression profiles separately for each type of uterine sarcoma - leiomyosarcoma, low-grade endometrial stromal sarcoma, high-grade endometrial stromal sarcoma (different subtypes if data is available), and undifferentiated uterine sarcoma. Again, adenosarcoma can be either removed or included as a separate entity - any relationship of miRNA expression with sarcomatous overgrowth and/or high-grade morphology? Also, for each sarcoma subtype it should be clearly described which miRNAs can be used for diagnosis and which are prognostic/predictive.
Thank you for this useful comment. We have now clearly indicated that paragraphs 2.1., 2.2., and 2.4., describe miRNA expression profiles for leiomyosarcomas. Paragraph 2.3. has now been subdivided into subparagraphs in order to separately describe miRNA expression profiles for each type of uterine sarcoma. In the Discussion part, we have now clearly described which miRNAs are characteristic (diagnosis, prognosis, prediction) of each sarcoma subtype.
Round 2
Reviewer 2 Report
Atypical leiomyoma is not a WHO category. Do the authors mean STUMP (smooth muscle tumor of uncertain malignant potential)? STUMP is not a subtype of uterine sarcoma. Therefore, any discussion of miRNA profiling in atypical leiomyoma / STUMP should be included in the differential diagnosis for leiomyosarcoma. 2.3.3. Adenosarcoma should be removed as there is only one paper.
Most of the miRNA data appears to be for leiomyosarcoma. Therefore the manuscript should focus on leiomyosarcoma and other uterine mesenchymal tumors should be included from the differential diagnosis point of view. The following changes for the subheadings are recommended:
2.1. Differential expression of miRNAs in uterine leiomyosarcoma cell lines
2.2. Different miRNA profiles between uterine leiomyosarcomas versus normal uteri and benign uterine tumors
2.3. Smooth muscle tumors of uncertain malignant potential versus leiomyosarcoma
2.4. Endometrial stromal sarcoma versus leiomyosarcoma
2.5. Undifferentiated pleomorphic sarcoma versus leiomyosarcoma
2.6. miRNAs correlate with prognosis of uterine leiomyosarcomas and could predict response to treatment
For each entity (e.g. endometrial stromal sarcoma, undifferentiated sarcoma, STUMP), please include a clear definition before discussing the miRNA profile.
Consider changing the title to "The Role of MicroRNAs in Uterine Leiomyosarcoma Diagnosis and Treatment."
See above
Author Response
Atypical leiomyoma is not a WHO category. Do the authors mean STUMP (smooth muscle tumor of uncertain malignant potential)? STUMP is not a subtype of uterine sarcoma. Therefore, any discussion of miRNA profiling in atypical leiomyoma / STUMP should be included in the differential diagnosis for leiomyosarcoma. 2.3.3. Adenosarcoma should be removed as there is only one paper.
In his work, Zhang et al. clearly focus their study results on atypical leiomyoma, which they define as one of the ‘‘at least 6 major histologically defined types: leiomyoma (ULM), mitotically active leiomyoma (MALM), cellular leiomyoma (CLM), atypical leiomyoma (ALM), uncertain malignant potential (STUMP), and leiomyosarcoma (LMS).’’ Following the reviewer’s suggestion, we have included any discussion of miRNA profiling in atypical leiomyoma in the differential diagnosis for leiomyosarcoma and removed adenosarcoma.
Most of the miRNA data appears to be for leiomyosarcoma. Therefore, the manuscript should focus on leiomyosarcoma and other uterine mesenchymal tumors should be included from the differential diagnosis point of view. The following changes for the subheadings are recommended:
2.1. Differential expression of miRNAs in uterine leiomyosarcoma cell lines
2.2. Different miRNA profiles between uterine leiomyosarcomas versus normal uteri and benign uterine tumors
2.3. Smooth muscle tumors of uncertain malignant potential versus leiomyosarcoma
2.4. Endometrial stromal sarcoma versus leiomyosarcoma
2.5. Undifferentiated pleomorphic sarcoma versus leiomyosarcoma
2.6. miRNAs correlate with prognosis of uterine leiomyosarcomas and could predict response to treatment
We have now changed the subheadings as proposed by the reviewer.
For each entity (e.g. endometrial stromal sarcoma, undifferentiated sarcoma, STUMP), please include a clear definition before discussing the miRNA profile.
We have now included a clear definition for each entity before discussing the miRNA profile.
Consider changing the title to "The Role of MicroRNAs in Uterine Leiomyosarcoma Diagnosis and Treatment."
We have now changed the title to "The Role of MicroRNAs in Uterine Leiomyosarcoma Diagnosis and Treatment."
Round 3
Reviewer 2 Report
Please rewrite the section about "atypical leiomyoma" as follows:
2.3. Other smooth muscle neoplasms versus leiomyosarcoma
In addition to leiomyosarcoma, smooth muscle neoplasms of the uterus include leiomyoma (with variants such as cellular leiomyoma, mitotically active leiomyoma, atypical or bizarre leiomyoma) and smooth muscle tumor of uncertain malignant potential (STUMP). Zhang et al. analyzed a cohort of 167 uterine smooth muscle tumors, including 38 leiomyosarcomas, 18 STUMPs, 42 bizarre leiomyomas, 22 cellular leiomyomas, 7 mitotically active leiomyomas, and 40 conventional leiomyomas. They found that bizarre leiomyomas and leiomyosarcomas share similar miRNA signatures [50].
See above
Author Response
Please rewrite the section about "atypical leiomyoma" as follows:
2.3. Other smooth muscle neoplasms versus leiomyosarcoma
In addition to leiomyosarcoma, smooth muscle neoplasms of the uterus include leiomyoma (with variants such as cellular leiomyoma, mitotically active leiomyoma, atypical or bizarre leiomyoma) and smooth muscle tumor of uncertain malignant potential (STUMP). Zhang et al. analyzed a cohort of 167 uterine smooth muscle tumors, including 38 leiomyosarcomas, 18 STUMPs, 42 bizarre leiomyomas, 22 cellular leiomyomas, 7 mitotically active leiomyomas, and 40 conventional leiomyomas. They found that bizarre leiomyomas and leiomyosarcomas share similar miRNA signatures [50].
We have now rewritten the section, as suggested by the reviewer.